# National health insurance membership in Indonesia: Do socio-economic elements matter?

**Misnaniarti**[1]*, **Wahyu Pudji Nugraheni**[2], **Agung Dwi Laksono**[2], **Asep Kusnali**[2], **Irfan Ardani**[2], **Leny Latifah**[2], **Rofingatul Mubasyiroh**[2], **Diah Yunitawati**[2], **Tati Suryati Warouw**[2]

**1** Faculty of Public Health, Sriwijaya University, Palembang, Indonesia, **2** National Research and Innovation Agency, Jakarta, Indonesia

\* misnaniarti@fkm.unsri.ac.id

## Abstract

The Indonesian government released a National Health Insurance (NHI) policy to realize universal health coverage (UHC) in Indonesia, yet disparities in membership persist. The study examined the influence of various socio-economic factors, including education, employment, and wealth status, on National Health Insurance (NHI) membership in Indonesia. Using a cross-sectional design, this study examined 1,223,377 individuals. The level of education was utilized as the exposure variable, while NHI membership served as the outcome variable. The analysis included residence, age, gender, marital status, and occupation as covariates. In the final stage, logistic multinomial regression was implemented. Based on the findings of this study, Indonesia's coverage for NHI membership in 2023 was 72.5%. Higher education levels are associated with increased likelihood of NHI participation, supporting the notion that education promotes greater access to Universal Health Coverage (UHC). Wealth status and demographic factors such as age and marital status also significantly associated with NHI membership. According to the results, the study concluded that education level and wealth status significantly influence NHI membership in Indonesia. Demographic factors also play essential roles. Tailored interventions addressing these factors are necessary to improve NHI enrollment and achieve universal health coverage.

## Introduction

Health insurance is a cornerstone of sustainable development, enhancing individual well-being and driving economic and social progress. Comprehensive programs have improved public health, reduced poverty, and promoted equitable access to essential services globally. In Indonesia, the National Health Insurance (NHI) program exemplifies this, addressing the unique challenges of a diverse population and evolving healthcare system. As a state initiative, the NHI has boosted economic growth and

**Data availability statement:** The data that supports the findings of this study are available from BPS-Statistics Indonesia, but restrictions apply to the availability of these data, which were used under license for the current study, and so are not publicly available. Data are however available from the corresponding author upon reasonable request and with permission of BPS-Statistics Indonesia. Requests for access to this data should be submitted to the Statistics Information Service System (https://silastik.bps.go.id/v3/index.php/site/login/; Email: pst@bps.go.id; Phone: +62(021) 350-7057).

**Funding:** The author(s) received no specific funding for this work.

**Competing interests:** The authors have declared that no competing interests exist.

healthcare access [1], reflecting the government's commitment to equitable healthcare [2]. Understanding the dynamics of this program within Indonesia contributes to the global discourse on UHC. Examining this program provides valuable lessons for strengthening health systems and advancing Universal Health Coverage (UHC) in similar settings.

As of December 2021, participation in the NHI had reached 235 million participants, covering approximately 86.55% of Indonesia's population (with a projected population of 272 million in 2021) [3]. However, this figure falls significantly short of the participation target set in the National Medium-Term Development Plan (RPJMN), which aims for 98% coverage by 2024 [4]. Fluctuations have been observed despite the overall increase in NHI membership since its inception in 2014. Membership decreased by 1.7 million in 2020, likely due to economic challenges, but rebounded with a 5.97% increase in 2021, reflecting the government's intensified efforts to expand coverage. These fluctuations highlight the challenges of sustaining membership growth and achieving the UHC target by 2024 [3].

This shortfall underscores governmental concerns regarding the suboptimal implementation of the compulsory enrollment principle to attain universal health coverage (UHC) under the NHI [5]. Various factors contribute to this shortfall, notably compliance enforcement, particularly among economically vulnerable demographics [6,7]. A comprehensive approach encompassing education, legal enforcement, and incentives is imperative to ensure a universal understanding of the significance and obligations associated with NHI participation [8,9].

Several factors influence NHI membership in Indonesia, including demographics (age, gender, and marital status) [10,11], socioeconomic conditions (wealth, education level) [12], and geographic disparities (urban vs. rural residence). Older individuals with stable employment and urban residents are more likely to own NHI due to better access to information, resources, and healthcare services [13,14]. Economic status also plays a crucial role, as higher-income individuals can afford premiums, while lower-income groups face affordability challenges. Additionally, individuals with chronic illnesses or poor self-perceived health are more inclined to own health insurance to support their medical needs. These factors, though significant, interact in complex ways, reflecting the multifaceted nature of NHI ownership in Indonesia. [11].

Education enhances health literacy, enabling individuals to understand the benefits of health insurance, navigate the enrollment process, and access healthcare services effectively. Education level has a significant impact on health insurance ownership. A study conducted in Indonesia found that pregnant women with higher education were 3.349 times more likely than those with no education to have health insurance [15]. Another study in China found that education significantly increased the demand for commercial health insurance [16]. Furthermore, research conducted on 26 OECD countries from 1995 to 2015 found that adults with higher educational attainment better understand and appreciate health insurance's benefits, thereby increasing ownership rates. Education improves access to resources, including health insurance,

which enables individuals to receive better healthcare services [17]. Additionally, educated individuals are more likely to engage in healthy behaviors and recognize the importance of health insurance.

UHC ensures that all individuals can access promotive, preventive, curative, and rehabilitative health services of sufficient quality without experiencing financial hardship [18]. In Indonesia, the government launched the NHI scheme under the National Social Security System Law 2004, which became operational in 2014, mandating universal participation. Despite a 95% coverage target by 2019, the scheme achieved 85.3% coverage by the end of that year [19]. The 2020–2024 RPJMN aims for 98% population participation in NHI to further UHC's goals of financial protection and equitable healthcare access. Presidential Instruction No. 1 of 2022 reinforces this effort, tasking all levels of government to accelerate UHC implementation, highlighting the shared responsibility of national and local governments in achieving this critical objective.

This study examines the role of various socio-economic elements—particularly education level, wealth status, employment, and residential location—in shaping NHI membership in Indonesia. These factors are especially relevant among disadvantaged populations facing compounded health insurance enrollment barriers. While previous studies have examined factors influencing NHI membership in Indonesia, limited research has explored how multiple socio-economic characteristics—such as education, employment, wealth, and geographic location—jointly influence NHI membership in Indonesia, particularly among the poor and disadvantaged groups. This research aims to assess multiple socio-economic factors, including education, that interact with structural disadvantages such as poverty and rural residence to influence NHI participation. The study further disaggregates the influence of education within subgroups to better understand its differential effect on advancing UHC. Studies in Ghana and India have demonstrated that higher education levels correlate with increased health insurance enrollment rates [20,21]. However, the specific dynamics in Indonesia, especially concerning the disadvantaged population, have not been fully explored. This study contributes to the literature by analyzing data from the 2023 National Socioeconomic Survey to examine these relationships.

The potential benefits derived from this research are substantial. Examining the role of education level on NHI membership in Indonesia can provide crucial insights that inform more targeted and effective policy making. Specifically, understanding how education influences health insurance uptake can help policymakers design interventions that better address the disparities in coverage, particularly in underserved regions like eastern Indonesia [22]. These insights could guide the allocation of resources, ensuring that investments in health infrastructure and human resources are directed where they are most needed, thereby reducing inequalities between provinces [23,24].

Moreover, this research has the potential to increase the overall effectiveness of the NHI Program by highlighting the importance of integrating educational initiatives with insurance enrollment efforts. This could lead to higher participation rates, especially among low-income and less-educated communities, ultimately contributing to greater health equity nationwide. By addressing these gaps, the study supports the broader goal of achieving universal health coverage in Indonesia, ensuring that every citizen, regardless of their educational background, can access the healthcare services they need.

## Materials and methods

### Data source

For this study, secondary data from the 2023 National Socioeconomic Survey were applied. Concurrently, Indonesian Statistics at the national level undertook a cross-sectional survey as part of the research. Data were collected for the survey throughout March 2023.

The 2023 National Socioeconomic Survey comprises the entirety of the Indonesian domestic population. The research encompasses 345,000 households dispersed across 514 districts/cities and 34 provinces within the Indonesian nation. A total of 34,500 census block samples were incorporated into the survey. Ten households were selected from each block

using systematic sampling, which was conducted using the Probability Proportional to Size (PPS) method. To ensure the sample's representativeness, stratification was executed at the household and census block levels within specific census blocks. Census blocks employ explicit stratification; for the 2020 Population Census, all inhabitants are categorized into regular census blocks according to their urban or rural status. On the contrary, implicit stratification is established based on the level of education the head of the household attained [25]. All Indonesian citizens residing in Indonesia, at least 15 years of age, were enrolled in this research. The sampling methods involved the selection of 1,223,377 participants as the sample.

## Setting

The study assessed NHI membership at the national level in Indonesia, focusing on understanding the socioeconomic factors influencing participation in the NHI scheme.

## Outcome variable

Membership in the NHI was included as a dependent variable in the research. The scope of the study encompassed all modes of NHI participation, including coverage obtained through alternative means, coverage provided by businesses or with government support, and mandatory enrollment for public servants, police, and army personnel. Additionally, "Yes" or "No" is selected as the survey response option regarding NHI membership.

## Exposure and socio-economic variables

Education level was utilized as an exposure variable in the investigation. The respondent's educational background is relevant in recognizing their most recent academic certification. The study examines four levels of education: elementary, junior high, senior high, and collegiate. Residences were classified into two categories: urban and rural. The survey employed the classification standards for urban and rural areas established by the Indonesian Central Bureau of Statistics. Individuals' employment status is classified into two categories: employed and unemployed. Participant revenue was ascertained through utilizing the wealth index formula in the survey. To determine the wealth index, the survey applied weights to the average of a household's expenditures. During the intervening period, the survey calculated the prosperity index by incorporating various components, including housing, food, healthcare, and essential household expenditures. Furthermore, the survey classified wealth status into five discrete categories: the richest, richer, middle, poor, and poorest [26].

## Control variables

The investigation employed six independent variables: education, residence type, age, gender, marital status, employment status, and wealth status. The age of the participants was ascertained using their most recent birthday. The survey delineated gender into two distinct categories, namely male and female. Furthermore, the survey categorised the participants' marital status into three discrete groups: those who had never been married or never in union, those who were still in marital relation or married, and those who were divorced or widowed.

## Data analysis

During the initial phases of the analysis, the researchers compared the dichotomous variable across two variables using the chi-squared test. Simultaneously, the researchers assessed the continuous age variable using the T-test. Additionally, a collinearity test was implemented in the study to determine whether or not the independent variables of the final regression model exhibited significant correlation. Logistic binary regression was applied to the analysis of the concluding results of the study. The study employed a method that had been previously devised to examine the multivariate correlation

between all independent variables and survey participation in the NHI [12]. The investigation performed the statistical analysis using IBM SPSS 26.

## Ethical approval

This investigation, utilizing secondary data, is exempt from review by the National Ethics Committee of the National Research and Innovation Agency.

## Results

Table 1 presents the outcome of descriptive statistics of participant analysis. Socioeconomic factors, including education, employment, wealth status, and residence, showed significant associations with NHI membership. For instance, individuals in urban areas and those in the wealthiest quintile were more likely to be enrolled in the NHI. According to residence type, urban areas have a higher ratio of NHI members than rural areas. Meanwhile, the study shows that the average age of people with NHI is older than those without.

**Table 1. Descriptive statistic of participants.**

| Demography Characteristics | NHI Membership | | | | | | p-value |
|---|---|---|---|---|---|---|---|
| | Total (n=1,223,377) | | No (n=336,682) | | Yes (n=886,695) | | |
| | n | % | n | % | n | % | |
| **Education level** | | | | | | | < 0.001 |
| Primary school | 681,164 | 55.7 | 213,739 | 31.4 | 467,425 | 68.6 | |
| Junior High School | 202,842 | 16.5 | 52,483 | 25.9 | 150,359 | 74.1 | |
| Senior High School | 249,249 | 20.4 | 56,787 | 22.8 | 192,462 | 77.2 | |
| College | 90,122 | 7.4 | 13,673 | 15.2 | 76,449 | 84.8 | |
| **Residence Type** | | | | | | | < 0.001 |
| Urban | 508,268 | 41.6 | 120,633 | 23.7 | 387,635 | 76.3 | |
| Rural | 715,109 | 58.4 | 216,049 | 30.2 | 499,060 | 69.8 | |
| **Age (mean)** | – | (31.9) | – | (28.1) | – | (33.4) | < 0.001 |
| **Gender** | | | | | | | |
| Male | 614,109 | 50.2 | 172,530 | 28.1 | 441,579 | 71.9 | |
| Female | 609,268 | 49.8 | 164,152 | 26.9 | 445,116 | 73.1 | |
| **Marital status** | | | | | | | < 0.001 |
| Never in union | 548,512 | 44.8 | 173,383 | 31.6 | 375,129 | 68.4 | |
| Married | 587,654 | 48.1 | 142,157 | 24.2 | 445,497 | 75.8 | |
| Divorced/Widowed | 87,211 | 7.1 | 21,142 | 24.2 | 66,069 | 75.8 | |
| **Employment Status** | | | | | | | < 0.001 |
| Unemployed | 697,644 | 57.0 | 208,833 | 29.9 | 488,811 | 70.1 | |
| Employed | 525,733 | 43.0 | 127,849 | 24.3 | 397,884 | 75.7 | |
| **Wealth Status** | | | | | | | < 0.001 |
| Poorest | 304,278 | 24.9 | 100,274 | 33.0 | 204,004 | 67.0 | |
| Poorer | 268,670 | 22.0 | 79,576 | 29.6 | 189,094 | 70.4 | |
| Middle | 243,694 | 19.9 | 65,763 | 27.0 | 177,931 | 73.0 | |
| Richer | 220,250 | 18.0 | 52,668 | 23.9 | 167,582 | 76.1 | |
| Richest | 186,485 | 15.2 | 38,401 | 20.6 | 148,084 | 79.4 | |

Based on gender, Table 1 displays that females have more NHI members than males. Regarding marital status, never in the union has the lowest ratio of NHI members. On the other hand, according to employment status, the employed have a higher NHI members ratio than the unemployed. Furthermore, the richest have the highest proportion of NHI members based on wealth status.

Following this, the collinearity examination was conducted. Based on the test results, no discernible correlation exists between the independent variables. The variance inflation factor value for each variable is less than 10.00, while the significance level of the tolerance value exceeds 0.10. During the investigation, it was determined that the regression model lacked multicollinearity; this indicates that the test's foundation for concluding was robust.

Table 2 presents National Health Insurance (NHI) membership distribution across various socioeconomic and demographic characteristics, stratified by education level. The data reveal a consistent trend in which individuals with higher educational attainment, particularly those with a college-level education, exhibit higher NHI membership rates across all categories, including urban residence, formal employment, marital status, and higher wealth quintiles. Urban residents, for instance, show a notably higher proportion of NHI membership as education level increases, with 85.97% of college-educated individuals insured, compared to 72.01% among those with only primary education. Similar patterns are observed among those in wealthier groups and those employed, suggesting that higher education correlates with improved access to resources that facilitate NHI enrollment. Statistically significant associations (p<0.001) and moderate effect sizes (Cramer's V ranging up to 0.128) underscore the multifactorial influences of education on insurance uptake in Indonesia.

Regarding NHI membership in Indonesia, the logistic binary regression analysis outcomes are shown in Table 3. Regarding education level, the result indicates that junior high school is 1.214 times more likely to be an NHI member than primary school 95% CI 1.200–1.228). Meanwhile, senior high school is 1.301 times more likely than primary school to

**Table 2. Distribution of NHI membership according to socioeconomic characteristics and education level in Indonesia (n=1,223,377).**

| Characteristics | NHI Membership (%) | | | | | | | | p-value | Cramer's V |
|---|---|---|---|---|---|---|---|---|---|---|
| | Yes | | | | No | | | | | |
| | Primary school | Junior high school | Senior high school | College | Primary school | Junior high school | Senior high school | College | | |
| **Residence Type** | | | | | | | | | | |
| Urban | 72.01 | 76.70 | 79.27 | 85.97 | 27.99 | 23.30 | 20.73 | 14.03 | <0.001 | 0.108 |
| Rural | 66.86 | 72.32 | 74.77 | 82.91 | 33.14 | 27.68 | 25.23 | 17.09 | <0.001 | 0.094 |
| **Gender** | | | | | | | | | | |
| Male | 68.06 | 73.38 | 76.68 | 84.03 | 31.94 | 26.62 | 23.32 | 15.97 | <0.001 | 0.107 |
| Female | 69.17 | 74.90 | 77.83 | 85.51 | 30.83 | 25.10 | 22.17 | 14.49 | <0.001 | 0.115 |
| **Marital status** | | | | | | | | | | |
| Never in union | 64.41 | 75.95 | 77.33 | 81.44 | 35.59 | 24.05 | 22.67 | 18.56 | <0.001 | 0.128 |
| Married | 73.68 | 72.61 | 77.12 | 85.75 | 26.32 | 27.39 | 22.88 | 14.25 | <0.001 | 0.092 |
| Divorced/ Widowed | 74.45 | 75.34 | 77.71 | 84.58 | 25.55 | 24.66 | 22.29 | 15.42 | <0.001 | 0.061 |
| **Work type** | | | | | | | | | | |
| No work | 66.59 | 75.56 | 78.11 | 83.28 | 33.41 | 24.44 | 21.89 | 16.72 | <0.001 | 0.114 |
| Work | 73.26 | 72.48 | 76.69 | 85.37 | 26.74 | 27.52 | 23.31 | 14.63 | <0.001 | 0.094 |
| **Wealth status** | | | | | | | | | | |
| Poorest | 64.17 | 72.35 | 73.25 | 76.55 | 35.83 | 27.65 | 26.75 | 23.45 | <0.001 | 0.089 |
| Poorer | 67.56 | 73.38 | 74.99 | 79.18 | 32.44 | 26.62 | 25.01 | 20.82 | <0.001 | 0.080 |
| Middle | 70.12 | 74.34 | 76.74 | 82.71 | 29.88 | 25.66 | 23.26 | 17.29 | <0.001 | 0.082 |
| Richer | 72.82 | 75.44 | 79.33 | 85.94 | 27.18 | 24.56 | 20.67 | 14.06 | <0.001 | 0.096 |
| Richest | 74.79 | 76.42 | 80.84 | 88.30 | 25.21 | 23.58 | 19.16 | 11.70 | <0.001 | 0.125 |

be an NHI member (95% CI 1.286-1-1.315). Moreover, college is 1.916 times more likely to own NHI than primary school (95% CI 1.878–1.954).

The six variables included in the study—age, gender, marital status, employment status, wealth status, and residence—were adopted as independent variables to predict NHI membership. Education was treated as the primary independent variable, while the others were controlled to isolate the effect of education on NHI membership. Analysis revealed significant differences in demographic and socioeconomic characteristics across education levels. Individuals with lower education levels (e.g., primary school) were more likely to reside in rural areas, have lower wealth status, and be employed in informal sectors. Conversely, those with higher education levels (e.g., college) predominantly lived in urban settings, exhibited higher wealth indices, and were more likely to be formally employed. Gender disparities were also noted; females were more represented among higher-educated groups than males.

According to residence type, someone in rural areas is 0.807 times less likely to be an NHI member than those in urban areas (95% CI 0.800–0.814). Moreover, the results show that four demographic characteristics are associated with NHI membership: age, gender, marital status, and employment status. Based on wealth status, Table 2 shows that the wealthier the status, the more likely the likelihood of being an NHI member.

## Discussion

Since 2014, the NHI program has been a cornerstone of the country's social health insurance system, aiming to ensure equitable access to healthcare for all citizens. Over the years, NHI coverage has steadily expanded, encompassing a growing number of participants across diverse socioeconomic groups [27]. By 2023, approximately 72.5% of the

**Table 3. The results of the logistic binary regression analysis concerning NHI membership in Indonesia in 2023 (n = 1,223,377).**

| Predictor | NHI Membership | | | |
| --- | --- | --- | --- | --- |
| | p-value | AOR | 95% CI | |
| | | | Lower Bound | Upper Bound |
| Education: Primary School (ref.) | – | – | – | – |
| Education: Junior High School | <0.001 | 1.214 | 1.200 | 1.228 |
| Education: Senior High School | <0.001 | 1.301 | 1.286 | 1.316 |
| Education: College | <0.001 | 1.916 | 1.878 | 1.954 |
| Residence: Urban (ref.) | – | – | – | – |
| Residence: Rural | <0.001 | 0.807 | 0.800 | 0.814 |
| Age | <0.001 | 0.644 | 0.642 | 0.645 |
| Gender: Male (ref.) | – | – | – | – |
| Gender: Female | <0.001 | 1.054 | 1.045 | 1.063 |
| Marital: Never in union (ref.) | – | – | – | – |
| Marital: Married | <0.001 | 1.264 | 1.251 | 1.276 |
| Marital: Divorced/Widowed | <0.001 | 1.305 | 1.283 | 1.328 |
| Employment: Unemployed (ref.) | – | – | – | – |
| Employment: Employed | <0.001 | 1.059 | 1.048 | 1.070 |
| Wealth: Poorest (ref.) | – | – | – | – |
| Wealth: Poorer | <0.001 | 1.114 | 1.102 | 1.127 |
| Wealth: Middle | <0.001 | 1.223 | 1.209 | 1.238 |
| Wealth: Richer | <0.001 | 1.371 | 1.354 | 1.389 |
| Wealth: Richest | <0.001 | 1.494 | 1.473 | 1.516 |

Note: AOR: Adjusted Odds Ratio; CI: confidence interval.

population was enrolled, though efforts are ongoing to reach remote and underserved areas. The program has also improved its benefits package, adding specialized treatments to address participants' evolving healthcare needs. During the COVID-19 pandemic, the NHI demonstrated resilience, providing essential services such as testing and treatment for COVID-19, reinforcing its role as a key pillar of Indonesia's healthcare system.

The government has prioritized digitalization efforts to enhance the efficiency and accessibility of the NHI program. Innovation includes developing integrated information systems for participant management, claims processing, and provider payments, enabling more streamlined and transparent operations [7,9]. Additionally, digital health platforms have been introduced to support telemedicine, simplify registration, and facilitate real-time data exchange between stakeholders. Efforts to improve the quality of services under the NHI program include implementing digital quality monitoring tools, establishing performance-based incentives, and utilizing analytics to identify service gaps [28]. However, challenges remain, such as ensuring equitable access to digital tools in remote areas, addressing data security concerns, and enhancing user adoption among healthcare providers and participants. By addressing these challenges and expanding digital innovations, the NHI program can better fulfill its goal of providing equitable and high-quality healthcare access for all citizens.

Education is critical in overcoming barriers to NHI enrollment, particularly for disadvantaged populations. Higher education levels enhance health literacy, enabling individuals to navigate complex enrollment processes and access healthcare services. Higher education levels enhance health literacy, enabling individuals to navigate complex enrollment processes and access healthcare services [14,29]. However, wealth and geographic disparities exacerbate challenges for low-income and rural populations [30]. These disparities can impact healthcare access equity. Policymakers should prioritize education-based interventions alongside financial subsidies to bridge these gaps and promote universal coverage.

Previous studies have found similar results, indicating that individuals with higher levels of education are more likely to participate in NHI programs, even within low-income groups [14]. This means that poor individuals with higher levels of education are more likely to enroll in NHI. This trend is often attributed to better awareness of healthcare options, a greater understanding of the benefits of health insurance, and potentially higher income levels are more common among those with higher education, all of which facilitate NHI enrollment [31–34].

The observed differences in characteristics between education levels suggest the need for tailored interventions. For instance, efforts targeting individuals with lower education levels should emphasize outreach in rural areas and focus on improving awareness through locally relevant health promotion campaigns. Financial barriers, common among lower-educated populations, could be addressed through subsidies or simplified registration processes. Meanwhile, interventions for higher-educated individuals could leverage digital platforms and workplace-based enrollment strategies to enhance NHI participation further. Recognizing these distinctions can help policymakers design more effective, education-specific strategies to achieve universal health coverage.

Higher education strongly correlates with increased NHI membership, as it enhances access to information, health literacy, and economic opportunities [35,36]. Studies from Indonesia [37] and other countries, including Ghana [21], India [20], and Congo [38], demonstrate that education improves employment prospects and income, enabling individuals to afford health insurance. Higher education levels in Indonesia are particularly linked to formal employment and economic security, underscoring the need for policies addressing educational disparities to promote equitable NHI enrollment [14,39].

According to residence type, people in rural areas were less likely to be NHI members than those in urban areas. Rural populations may have lower health literacy levels and less awareness of the importance of health insurance. Limited access to information and educational resources about NHI programs can contribute to lower enrollment rates in rural areas [40]. Additionally, Indonesia's status as a middle-income country with a population of 262 million dispersed across 17,744 islands provides special challenges for the health system and UHC coverage [7]. The leading cause of low health insurance coverage in rural Indonesia is a lack of public information, socialization, education, and health promotion media, particularly about insurance membership and how to obtain it [41].

This contributes to household leaders' limited awareness of the need for health insurance protection in supporting productivity. Insurance knowledge is essential to increasing premium payment compliance and expanding insurance coverage [42].

Socioeconomically disadvantaged communities in rural areas of Indonesia face significant barriers to accessing primary healthcare, which impacts their participation in the NHI program. These barriers include a shortage of healthcare facilities and providers and logistical challenges like long-distance and inadequate transportation options, making it difficult for rural residents to enroll in NHI, even if they understand its benefits [43]. Additionally, cultural beliefs that emphasize self-reliance and the use of home remedies, coupled with skepticism toward government institutions, further diminish the perceived need for health insurance in these communities. These factors contribute to lower NHI enrollment rates in rural areas, consistent with findings from a meta-analysis of 12 publications, which identified rural residency as a significant factor influencing NHI ownership in Indonesia.

Moreover, the results show that four demographic characteristics are associated with NHI membership: age, gender, marital status, and employment status. Younger individuals are less likely to enroll in NHI due to limited awareness of the importance of health insurance or financial constraints. Conversely, older adults, especially those nearing retirement age or experiencing health issues associated with ageing, are more likely to prioritize health insurance coverage and thus have higher enrollment rates [44]. Gender can influence NHI membership rates due to differences in healthcare-seeking behaviors and employment patterns [34]. Women, particularly those of childbearing age, may be more likely to enroll in NHI due to pregnancy-related healthcare needs and preventive care services. Men may be less likely to prioritize healthcare or may face barriers to enrollment due to factors such as employment instability or lower health literacy levels. Married individuals, especially those with children, may prioritize health insurance coverage for their families, leading to higher enrollment rates. Single individuals, particularly those without dependents, may be less likely to prioritize health insurance or face financial barriers to enrollment. Employment status is a significant determinant of NHI membership, as many countries tie health insurance coverage to employment. Full-time employees often have access to employer-sponsored health insurance plans or may be required by law to enroll in NHI, leading to higher enrollment rates among this group [44].

Based on wealth status, the wealthier the status, the more likely the likelihood of being an NHI member. The association between wealthier socioeconomic status and higher NHI membership rates reflects the intersection of financial resources, access to employer-sponsored benefits, healthcare utilization behaviors, health literacy, and perceptions of healthcare as a priority [45]. Wealthier individuals typically have more financial resources available to afford health insurance premiums. NHI programs often require individuals to pay premiums or contributions, and those with higher incomes are better positioned to meet these financial obligations without significant strain on their budgets. Even with insurance coverage, individuals may still face out-of-pocket costs such as deductibles, copayments, and coinsurance [44]. Wealthier individuals can better afford these costs, reducing financial barriers to seeking healthcare services and thus increasing the likelihood of NHI membership.

## Strength and limitation

The information utilized in this investigation was analyzed from a previous cross-sectional survey. The authors could not verify the temporal correlation between the exposure variable and the outcome. Several well-established variables, including income, smoking habit, and history of chronic illness, which were identified in other studies examining the determinants of health insurance ownership, were not considered in this investigation [46–48].

This study analyzed overall NHI membership without distinguishing between different types of insurance schemes, such as government-subsidized programs for low-income groups, employer-sponsored plans, or independent enrollment. This limitation restricts the ability to draw nuanced conclusions about how specific insurance types of impact membership rates across various socioeconomic groups.

## Conclusions

This study demonstrates that educational attainment plays is pivotal in determining enrollment in Indonesia's National Health Insurance (NHI) program. Individuals with higher levels of education are significantly more likely to be NHI members, which may be attributed to greater health literacy, better understanding of insurance benefits, and enhanced access to relevant information. In addition to education, other socio-economic and demographic variables—including wealth status, residence (urban or rural), age, sex, marital status, and employment—were also found to be statistically significant predictors of NHI membership.

The findings indicate that individuals residing in rural areas, those with lower educational levels, the unemployed, and members of the lowest wealth quintiles are less likely to be enrolled in the NHI. This underscores persistent disparities in access to social protection and suggests the need for more equitable outreach and policy intervention.

To improve NHI coverage and ensure that the goal of universal health coverage (UHC) is realized, particular attention must be directed towards these underserved populations. Policy responses should include strengthening community-based health literacy programs, simplifying administrative enrollment procedures, offering premium waivers or targeted subsidies for low-income groups, and enhancing the accessibility of primary healthcare services across all regions, particularly in rural and remote areas.

## Acknowledgments

The author expresses gratitude to Indonesian Statistics for granting access to the 2023 National Socioeconomic Survey so that this research can be conducted.

## Author contributions

**Conceptualization:** Misnaniarti Misnaniarti, Wahyu Pudji Nugraheni, Agung Dwi Laksono.

**Data curation:** Asep Kusnali, Irfan Ardani.

**Formal analysis:** Misnaniarti Misnaniarti, Wahyu Pudji Nugraheni, Irfan Ardani, Leny Latifah, Tati Suryati Warouw.

**Methodology:** Misnaniarti Misnaniarti, Wahyu Pudji Nugraheni, Agung Dwi Laksono, Asep Kusnali.

**Project administration:** Asep Kusnali, Diah Yunitawati.

**Resources:** Rofingatul Mubasyiroh.

**Software:** Leny Latifah, Rofingatul Mubasyiroh.

**Supervision:** Wahyu Pudji Nugraheni.

**Validation:** Irfan Ardani, Leny Latifah.

**Visualization:** Diah Yunitawati.

**Writing – original draft:** Agung Dwi Laksono, Tati Suryati Warouw.

**Writing – review & editing:** Misnaniarti Misnaniarti, Asep Kusnali, Irfan Ardani, Diah Yunitawati.

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
