## [Decision Letter · Decision Letter 0]

23 Jul 2024

PONE-D-24-21104National health insurance membership in Indonesia: Does education level matter?PLOS ONE

Dear Dr. Kusnali,

Thank you for submitting your manuscript to PLOS ONE. After careful consideration, we feel that it has merit but does not fully meet PLOS ONE’s publication criteria as it currently stands. Therefore, we invite you to submit a revised version of the manuscript that addresses the points raised during the review process.

We look forward to receiving your revised manuscript.

Kind regards,

Isaac Akintoyese Oyekola

Academic Editor

PLOS ONE

2. PLOS requires an ORCID iD for the corresponding author in Editorial Manager on papers submitted after December 6th, 2016. Please ensure that you have an ORCID iD and that it is validated in Editorial Manager. To do this, go to ‘Update my Information’ (in the upper left-hand corner of the main menu), and click on the Fetch/Validate link next to the ORCID field. This will take you to the ORCID site and allow you to create a new iD or authenticate a pre-existing iD in Editorial Manager. Please see the following video for instructions on linking an ORCID iD to your Editorial Manager account: "https://www.youtube.com/watch?v=_xcclfuvtxQ".

4. In the online submission form, you indicated that [The data that support the findings of this study are available from Statistics Indonesia, but restrictions apply to the availability of these data, which were used under license for the current study, and so are not publicly available. Data are however available from the corresponding author upon reasonable request and with permission of Statistics Indonesia.]. 

Additional Editor Comments:

Definitely, the paper investigated beyond education. Hence, and as contained in the covering letter, “National health insurance membership in Indonesia: Do socio-economic elements matter” may be considered more appropriate for the title.

In addition to reviewers’ comments, authors should note the following while revising the manuscript:

• Authors should state the research problem and demonstrate the uniqueness of this study. Generally, the background and discussion sections should be more robust, demonstrating scholarship

• The first acronym in Introduction, ‘NHIP’ was not first defined.

• “The survey results indicated that the furnished residences belonged to two discrete classifications: urban and rural.” Why reporting results in methodology? Which one is furnished residences? I believe authors meant “residence was classified into two categories: urban and rural” OR “residence had two categories: Urban and rural”. Authors must carefully select every word used. Just like, “The study displayed that the better the education, the higher the possibility of being an NHI member.” Which one is better education? ALSO, “During this period, individuals' employment status is classified into two separate categories: employed and indigent.” This is not a good classification for employment status. Seeing Table 1, the authors meant to classify into ‘employed and unemployed’. LIKEWISE, “survey classified the income index into five discrete categories, namely the most prosperous, indigent, middle-class, and poorest [23].” Again, this categorization is misleading. Indigent and poorest? Four classifications? Authors must correct this based on the information in Table 1.

• “Domicile, age, gender, marital status, employment status, and wealth status comprised the seven criteria”. Education is missing here, although authors may claim it was an exposure variable. Also, domicile might not be much appropriate, but residence.

• “During the initial phases of the sample”. Of the analysis, or sample?

• “Meanwhile, Fig 1 shows a map illustrating the geographical distribution of hospital utilization across the provinces of Indonesia in 2023. The map indicates that the lowest proportion of NHI members is in Kalimantan and the Nusa Tenggara region.” based on the statement, The Map in Figure 1 is not informative, and may not be considered relevant. Also, the map could not be accessed via the link provided.

• “The investigation determined that Indonesia's average NHI membership rate will be 72.5% in 2023 [Why not, ‘Indonesia’s average NHI membership rate was projected in 20XX to be 72.5% by 2023’]. Meanwhile, Fig 1 shows a map illustrating the geographical distribution of hospital utilization across the provinces of Indonesia in 2023. The map indicates that the lowest proportion of NHI members is in Kalimantan and the Nusa Tenggara region.” The statements did not seem to be the outcome of the investigation. Even the Map was not referenced. These statements would have been better in either Introduction or Discussions.

• Why starting to report Table 1 from gender. Interpret either ‘according to variables’ or ‘base on highest enrolment’.

• What was the reference category for age in Table 2? There is need to show the age category in Table 1, as this will give reader detailed information. The authors should report relevant statistics such as R-square, Coefficient and Standard Errors in Table 2.

• Authors may consider comparing between years (since such cross sectional study is conducted periodically). This is expected to improve the robustness of the study.

• Authors must report the sample size (n) of each category of socio-economic elements (putting % in bracket) in Table 1.

• The note ‘∗p < 0.001’ might not be relevant since no reference to such in the Table.

• “The study informs that six control variables are related to NHI membership in Indonesia.” Were these not the adopted variables in the study. Were other unrelated variables identified? In fact, the Table 2 would not have informed/confirmed the relationship that existed among variables (although it predicted what happens to dependent/outcome variable). Correlation and Chi-Square are necessary to show level and significance of relationship.

• There might be need to report cross tabulation and Cramer’s V to show the relationship and determine the level of corrections.

Discussion

• “Based on Ministry of Health data, the NHI membership coverage is about 95% in 2023”. Compare to “the investigation determined that Indonesia's average NHI membership rate was 72.5% in 2023”. Clarity is required since the study used the same year 2023 survey data.

• Education level plays a significant role in determining NHI membership among the poor population in Indonesia [12,26]. Not only education. Even, the statement is not novel.

• “Moreover, the results show that four [additional] demographic characteristics are associated with NHI membership”. Wealth index/status is missing here, although still discussed in last paragraph.

• The manuscript will benefit from improved discussions.

• Recommendations did not emanate from results. This is very important in scholarly research. For examples, “Continuously expand the coverage of the NHI to include more citizens, especially those from low-income backgrounds and remote areas.” AND “Conduct public awareness campaigns to educate citizens about the benefits of NHI membership, how to enroll, and the services available under the scheme” did not come from the results.” AND “Ensure that healthcare services covered under the NHI meet quality standards.”

ERRORS

• “This investigation utilizing secondary data is exempt from review by the National Ethics Committee of the National Research and Innovation Agency.”

• “qualitative study revealed that Information about plans, services, benefits, and opportunities to get health insurance are important factors in insurance enrolment [34]”. WHY USING UPPER CASE FOR ‘Information’

• “The relationship between higher education and higher enrolment in health insurance is similar to research in Ghana and Thailand [35,36].” authors must have mistaken India for Thailand

• Younger individuals, particularly those who are students or starting their careers, may be less likely to enroll in NHI due to limited awareness of the importance of health insurance or financial constraints. No longer ‘may’. Also authors may remove ‘particularly those who are students or starting their careers’ since age is the focus here not education.

CITATIONS REQUIRED

• “The study employed a method that had been previously devised to examine the multivariate correlation between all independent variables and survey participation in the NHI.” Citation is required.

• Some previous studies showed the same results that Higher education is associated with a higher likelihood of NHI membership among the poor. CITATION IS REQUIRED

• Authors must cite all substantive statements. For examples, out of many in Introduction, Methodology, and Discussion, “Another study showed that Indonesia's health insurance system covers many hospitalized patients, especially the poorest.” AND “However, the eastern region of Indonesia still lacks access to many services, and their benefits are not fully realized.” AND “Demographic factors such as age, region, residence, education, marital status, and employment status play a significant role in the ownership of National Health Insurance (NHI) in Indonesia. For instance, older individuals or those with stable employment are more likely to own NHI.”

• Authors must include complete references such as inclusion of doi.

Reviewers' comments:

Reviewer's Responses to Questions

**Comments to the Author**

1. Is the manuscript technically sound, and do the data support the conclusions?

Reviewer #1: Partly

Reviewer #2: Yes

Reviewer #3: Yes

2. Has the statistical analysis been performed appropriately and rigorously? 

Reviewer #1: No

Reviewer #2: Yes

Reviewer #3: Yes

3. Have the authors made all data underlying the findings in their manuscript fully available?

Reviewer #1: Yes

Reviewer #2: Yes

Reviewer #3: Yes

4. Is the manuscript presented in an intelligible fashion and written in standard English?

Reviewer #1: Yes

Reviewer #2: Yes

Reviewer #3: Yes

5. Review Comments to the Author

Reviewer #1: In my opinion, the relationship between education level and insurance membership is not a meaningful research topic worth pursuing as there is no evidence or knowledge gap for this issue. The author provided a comprehensive summary on the factors that impacted health insurance membership in Introduction and it is clear that there is no new information in this study that can contribute to literature.

Reviewer #2: Introduction

Improve the first paragraph by establishing a context that emphasizes the importance of discussing health insurance issues from a global perspective. Then, continue with a more specific explanation regarding the reasons why research on health insurance in Indonesia is very important.

Following the statement of purpose, it is important to highlight the potential benefits that can be derived from the results of this research. These benefits can range from informing policymaking to increasing the effectiveness of health insurance programs.

Methods and results

The outcome variable in this research is membership in Indonesia's national health insurance. As is known, national health insurance membership in Indonesia is divided into several types, consisting of Contribution Assistance Recipients (PBI in Indonesian), Wage Recipient Workers (PPU), Non-Wage Recipient Workers (PBPU in Indonesian), and Non-Workers (PB in Indonesian). Why doesn't the author differentiate National Health Insurance membership based on this scheme? Can data analysis be carried out taking into account the existence of several of these schemes? If the author has reasons, explain clearly why the data analysis design ignores the types of JKN participation mentioned above.

Discussion

In line 266, the author states the relationship between educational-level access to information and health literacy. Apart from these two things, there is a possibility that education is also related to work and welfare, as quoted from research in India. What about research in Indonesia? Add discussion about this.

Reviewer #3: PLOS ONE National health insurance membership in Indonesia: Does education level matter? PONE-D-24-21104

General comment:

The authors examined the role of education level on NHI membership in Indonesia. Using a cross-sectional design, this study examined 1,223,377 individuals using the latest secondary data from the 2023 National Socioeconomic Survey showing fluctuations in NHI participation since 2021. Authors provided justification the reason of fluctuations indicating Various factors contribute to this shortfall, notably compliance enforcement, particularly among economically vulnerable demographics. A comprehensive approach encompassing education, legal enforcement, and incentives is imperative to ensure a universal understanding of the significance and obligations associated with NHIP participation.

Specific comments

1. Authors focus of interest was contribution of education level to public awareness to participate in NHI membership served as the outcome variable

Comment: the article focused on contribution of education level on public awareness to participate in NHI independently and voluntarily. Please elaborate what the authors mean with public awareness. Is it in the level of awareness or the level of participation. Authors need to justify choosing level of education on NHI participation as variable of interest. As most study identified those with high education level has better health status compared to those with lower education. The authors pointed out that wealthier group have greater chance to be NHI member, which is common in most countries that those with better education and economic status is more concerned on their health, including NHI participation. What is the specific condition of Indonesian NHI compared to NHI in other countries which make the authors pay specific interest on analysis contribution of education level on NHI participation.

"The authors should clarify if they are examining the contribution of education level to public awareness of NHI benefits, or actual participation in the program."

2. Authors admit that Education plays a crucial role in influencing health insurance ownership and utilization. Education level has a significant impact on health insurance ownership. A study conducted in Indonesia found that pregnant women with higher education were 3.349 times more likely than 86 those with no education to have health insurance. Another study in China found that educations significantly increased the demand for commercial health insurance. Furthermore, research conducted on 26 OECD countries from 1995 to 2015 found that adults with higher educational attainment better understand and appreciate health insurance's benefits, thereby increasing ownership rates. Education improves access to resources, including health insurance, which enables individuals to receive better healthcare services. Additionally, educated individuals are more likely to engage in healthy behaviors and recognize the importance of health insurance.

Since contribution of education on NHI membership is the authors main interest, please provide more justification why selecting the variable.

Authors also admit the contribution of other demographic factors on NHI participation. Please elaborate how the intertwined of education level with other demographic factors on NHI participation.

"While the link between education and health insurance is established, the authors should elaborate on why education is a specific interest here. Do they expect a different dynamic in Indonesia's NHI compared to other countries?"

3. Table 1: please add total N in the left column of NHI membership, with column percent to make reader understand the big picture of population according to each explanatory variable.

4. How is the strength of the model as shown in the classification table? How the authors explain the strength of the model to increase NHI participation level?

5. Line 192: authors gave figure on the hospital utilisation. Please explain how the association of the figure with the topic of the manuscript which is contribution of education on NHI membership.

"It would be helpful to understand how the hospital utilization figure connects to the main topic of education and NHI participation. Is it presented as a potential benefit of NHI membership?"

6. Line 258 – 260: please elaborate the meaning of the sentence some previous studies showed the same results that Higher education is associated with a higher likelihood of NHI membership among the poor. Did authors mean poor people with higher education more likely to participate in NHI?

"This sentence could be rephrased for clarity. Did the authors find that even among low-income individuals, those with higher education were more likely to participate in NHI?"

7. The analysis showed that higher education and wealthiest are more likely to participate on NHI than those with lower education and least wealthy. How is the proportion of higher education and wealthiest among Indonesian compared to those with lower education and least wealthy? As authors said those with higher education and wealthier is more exposed to benefit of NHI. What authors recommendation for the government for people with lower education, least wealthy, less exposure to get NHI benefit.

8. Line 289 – 293: Socioeconomically disadvantaged communities living in rural areas in Indonesia face a variety of barriers that limit their access to primary health care; this is the government's primary concern, including the availability of a sufficient number and quality of health workers, combined with a telemedicine approach, to ensure the availability of health services and a reference guide:

Comment: what authors intend to say Socioeconomically disadvantaged communities living in rural areas in Indonesia face a variety of barriers that limit their access to primary health care with the telemedicine approach? Did authors intend to say telemedicine approach as solution for the socioeconomically disadvantaged communities living in rural areas?

o "The authors' discussion of telemedicine as a solution for rural communities with NHI access limitations needs clarification. Is telemedicine presented as a potential solution to improve access for these communities?"

9. Line 294 – 295: did author mean living in rural as barrier for NHI participation?

10. Authors provided comprehensive explanation on the dynamics of NHI participation

11. Thank you

6. PLOS authors have the option to publish the peer review history of their article (what does this mean? ). If published, this will include your full peer review and any attached files.

**Do you want your identity to be public for this peer review?** For information about this choice, including consent withdrawal, please see our Privacy Policy .

Reviewer #1: No

Reviewer #2: **Yes: ** Ratna Dwi Wulandari

Reviewer #3: **Yes: ** siti isfandari

---

## [Author Response · Author response to Decision Letter 1]

28 Sep 2024

Dear Editor,

Thank you for your email and the opportunity to revise our manuscript, which was submitted to PLOS ONE. We appreciate the constructive feedback provided by the reviewers and the academic editor.

I am pleased to inform you that we have completed the revisions based on the reviewers' comments. We have addressed each point thoroughly and made the necessary improvements to the manuscript. I will be submitting the revised version, along with the response to the reviewers, shortly.

Thank you once again for your consideration. I look forward to the next steps in the review process.

Best regards,

Misnaniarti

---

## [Decision Letter · Decision Letter 1]

3 Nov 2024

PONE-D-24-21104R1National health insurance membership in Indonesia: Does education level matter?PLOS ONE

Dear Dr. Misnaniarti,

Thank you for submitting your manuscript to PLOS ONE. After careful consideration, we feel that it has merit but does not fully meet PLOS ONE’s publication criteria as it currently stands. Therefore, we invite you to submit a revised version of the manuscript that addresses the points raised during the review process.

We look forward to receiving your revised manuscript.

Kind regards,

Isaac Akintoyese Oyekola

Academic Editor

PLOS ONE

Journal Requirements:

Additional Editor Comments:

The authors have been able to attend to the comments well, except for the following minor corrections:

‘National health insurance membership in Indonesia: Do socio-economic elements matter’ may be considered more appropriate for the manuscript considering the employment of socio-economic variables such as education, wealth, et cetera (NOT ONLY EDUCATION, ALTHOUGH EDUCATION MAY HAVE ‘DOMINANT INFLUENCE’).

Conclusions and recommendations must emanate from findings. For examples, “…COMMUNITY WELFARE provide more opportunities for individuals to engage in NHI” AND “…targeted health literacy initiatives should focus on INFORMAL SECTOR WORKERS…” were not part of the findings. Many more conclusions can be drawn from the findings and more recommendations can be made without exaggeration. Authors should maximize the use of this section to benefit readers and policy makers in a concise manner. For instance, since wealth influenced NHI membership, subsidy/premium waiver for the less-privileged may be more appropriate as recommendation, among others.

Reviewers' comments:

Reviewer's Responses to Questions

**Comments to the Author**

1. If the authors have adequately addressed your comments raised in a previous round of review and you feel that this manuscript is now acceptable for publication, you may indicate that here to bypass the “Comments to the Author” section, enter your conflict of interest statement in the “Confidential to Editor” section, and submit your "Accept" recommendation.

Reviewer #2: All comments have been addressed

Reviewer #3: (No Response)

2. Is the manuscript technically sound, and do the data support the conclusions?

Reviewer #2: Yes

Reviewer #3: Yes

3. Has the statistical analysis been performed appropriately and rigorously? 

Reviewer #2: Yes

Reviewer #3: Yes

4. Have the authors made all data underlying the findings in their manuscript fully available?

Reviewer #2: Yes

Reviewer #3: Yes

5. Is the manuscript presented in an intelligible fashion and written in standard English?

Reviewer #2: Yes

Reviewer #3: No

6. Review Comments to the Author

Reviewer #2: The author has responded well to all my previous comments in this revised manuscript.

However, my previous comment about the analysis based on the type of insurance may need to be mentioned in the limitations or added to the suggestions for further researchers to conduct a more in-depth study related to this.

Another crucial point that needs clarification is the need for more detailed information about the differences in characteristics between levels of education. This information is vital for designing effective interventions at each level of education, as the manuscript's title suggests a focus on education.

Reviewer #3: *Comment:**

Thank you for your excellent work in addressing and clarifying the reviewer inputs. Below are my specific comments:

**Abstract:**

- Line 30–31: Consider replacing "control variables" with "covariates."

- Line 39–41: Why do the authors only mention wealth and education in the conclusion?

**Introduction:**

- Line 45–59: The paragraph is too long and includes only two citations. Please condense and shorten it.

- Line 60–73: The main point is the fluctuation pattern of NHI membership. Please clarify the sentences to reflect this.

- Line 74–92: The authors discuss the contributing factors to NHI membership. Please streamline the paragraph to make it more concise and systematic.

- Line 100: Suggest emphasizing that education improves health literacy, leading to better access to health services.

- Line 104–117: This paragraph is too lengthy with only one citation. Please shorten it.

- Line 118–126: The authors seem particularly interested in the role of education in NHI membership among disadvantaged groups. It may be better to focus the analysis on this issue, comparing the impact of education on NHI membership between disadvantaged and better-off groups.

**Results:**

- Line 224: What are the authors trying to convey with Table 2? Please clarify the intent.

- Line 233: In Table 3, the authors should consider presenting the varying contribution of education to NHI membership across different wealth statuses.

**Discussion:**

- Line 247–259: This paragraph is too long and contains only one citation. Please condense it and highlight key findings from your analysis in the first paragraph.

- Line 260–269: The authors discuss the government program, its efforts, and challenges. What innovations are already in place, and what challenges should the government address through digitalization?

- Line 270–284: What are the key points the authors intend to emphasize?

- Line 285–303: What is the main takeaway from this paragraph? Please make it concise and clearer.

Great job overall!

7. PLOS authors have the option to publish the peer review history of their article (what does this mean? ). If published, this will include your full peer review and any attached files.

**Do you want your identity to be public for this peer review?** For information about this choice, including consent withdrawal, please see our Privacy Policy .

Reviewer #2: **Yes: ** Ratna Dwi Wulandari

Reviewer #3: **Yes: ** Siti Isfandari

---

## [Author Response · Author response to Decision Letter 2]

15 Jan 2025

Dear Reviewer and Editor,

Thank you for the opportunity to revise our manuscript. We appreciate the detailed feedback provided by the reviewers and the editorial team. We have carefully addressed each comment and made the necessary revisions to improve the clarity, depth, and overall quality of the manuscript.

As requested, we are submitting the following materials:

1. A detailed rebuttal letter responding to each comment raised by the reviewers and editor, outlining the changes made to the manuscript.

2. A marked-up copy of the revised manuscript highlighting all modifications using tracked changes.

3. A clean version of the revised manuscript without tracked changes.

We believe the revisions have strengthened the manuscript and hope it now meets PLOS ONE’s publication criteria. If further adjustments are required, we are more than willing to address them promptly.

Thank you for your guidance throughout the process.

Sincerely,

Dr. Misnaniarti

---

## [Decision Letter · Decision Letter 2]

24 Mar 2025

PONE-D-24-21104R2National health insurance membership in Indonesia: Do socio-economic elements matter?PLOS ONE

Dear Dr. Misnaniarti,

Thank you for submitting your manuscript to PLOS ONE. After careful consideration, we feel that it has merit but does not fully meet PLOS ONE’s publication criteria as it currently stands. Therefore, we invite you to submit a revised version of the manuscript that addresses the points raised during the review process.

We look forward to receiving your revised manuscript.

Kind regards,

Isaac Akintoyese Oyekola

Academic Editor

PLOS ONE

Journal Requirements:

Additional Editor Comments:

The authors have been able to attend to the comments well in ‘response to editor’s comment section’. However, I was wondering if the authors effected the corrections in the manuscript submitted for review. Specifically, there seems not to be any difference between the conclusion in Revision 2 and Revision 1.

Additionally, and as noted by the reviewer, authors must modify all sections in the manuscript to reflect the new title.

Reviewers' comments:

Reviewer's Responses to Questions

**Comments to the Author**

1. If the authors have adequately addressed your comments raised in a previous round of review and you feel that this manuscript is now acceptable for publication, you may indicate that here to bypass the “Comments to the Author” section, enter your conflict of interest statement in the “Confidential to Editor” section, and submit your "Accept" recommendation.

Reviewer #3: All comments have been addressed

2. Is the manuscript technically sound, and do the data support the conclusions?

Reviewer #3: Yes

3. Has the statistical analysis been performed appropriately and rigorously? 

Reviewer #3: Yes

4. Have the authors made all data underlying the findings in their manuscript fully available?

Reviewer #3: Yes

5. Is the manuscript presented in an intelligible fashion and written in standard English?

Reviewer #3: Yes

6. Review Comments to the Author

Reviewer #3: generally good. however authors need to correct statements in abstract and introduction in the objective and aim of the study

7. PLOS authors have the option to publish the peer review history of their article (what does this mean? ). If published, this will include your full peer review and any attached files.

**Do you want your identity to be public for this peer review?** For information about this choice, including consent withdrawal, please see our Privacy Policy .

Reviewer #3: **Yes: ** Siti Isfandari MA

---

## [Author Response · Author response to Decision Letter 3]

4 May 2025

We sincerely thank the editor and reviewer for their constructive feedback. In response to the general comment regarding the abstract and title, we have revised the abstract to better reflect the broader objective of examining the influence of multiple socio-economic factors, including but not limited to education level, on NHI membership. Regarding the objective statement, we clarified in both the abstract and introduction that the study aims to examine how education level, along with other socio-economic variables, affects NHI membership, with particular attention to disparities across economic strata. Finally, we have revised the abstract sentences for clarity and conciseness, as suggested.

---

## [Decision Letter · Decision Letter 3]

19 May 2025

PONE-D-24-21104R3National health insurance membership in Indonesia: Do socio-economic elements matter?PLOS ONE

Dear Dr. Misnaniarti,

Thank you for submitting your manuscript to PLOS ONE. After careful consideration, we feel that it has merit but does not fully meet PLOS ONE’s publication criteria as it currently stands. Therefore, we invite you to submit a revised version of the manuscript that addresses the points raised during the review process.

We look forward to receiving your revised manuscript.

Kind regards,

Isaac Akintoyese Oyekola

Academic Editor

PLOS ONE

Journal Requirements:

Additional Editor Comments:

* Complete title of Table 2, if still relevant.

* Focusing on socio-economic elements, and not only on education, might address the reviewer’s comments.

* Include age categories also (not only mean age) and report the statistics accordingly in Tables 1, 2, and 3.

Reviewers' comments:

Reviewer's Responses to Questions

**Comments to the Author**

1. If the authors have adequately addressed your comments raised in a previous round of review and you feel that this manuscript is now acceptable for publication, you may indicate that here to bypass the “Comments to the Author” section, enter your conflict of interest statement in the “Confidential to Editor” section, and submit your "Accept" recommendation.

Reviewer #3: All comments have been addressed

2. Is the manuscript technically sound, and do the data support the conclusions?

Reviewer #3: Yes

3. Has the statistical analysis been performed appropriately and rigorously? 

Reviewer #3: Yes

4. Have the authors made all data underlying the findings in their manuscript fully available?

Reviewer #3: Yes

5. Is the manuscript presented in an intelligible fashion and written in standard English?

Reviewer #3: Yes

6. Review Comments to the Author

Reviewer #3: please perform statstic analysis in line with the objective which was to identify the contribution of education level on UHC participation among poor and disadvantage group

7. PLOS authors have the option to publish the peer review history of their article (what does this mean? ). If published, this will include your full peer review and any attached files.

**Do you want your identity to be public for this peer review?** For information about this choice, including consent withdrawal, please see our Privacy Policy .

Reviewer #3: **Yes: ** Siti Isfandari

---

## [Author Response · Author response to Decision Letter 4]

4 Jul 2025

Thank you for the opportunity to revise our manuscript submitted to PLOS ONE. We appreciate the constructive feedback provided by you and the reviewers, which has helped us improve the quality and clarity of our work. We have carefully considered all comments and revised the manuscript accordingly.

We have provided a detailed point-by-point response to each comment raised by the reviewers and the academic editor. Changes made in the manuscript are marked using track changes (see the file titled Revised Manuscript with Track Changes). A clean version is also submitted as required.

We hope that our revised manuscript and responses adequately address the reviewers' and editor’s concerns. We thank you once again for the opportunity to improve our work and look forward to your further consideration.

Sincerely,

Dr. Misnaniarti

---

## [Decision Letter · Decision Letter 4]

22 Jul 2025

National health insurance membership in Indonesia: Do socio-economic elements matter?

PONE-D-24-21104R4

Dear Dr. Misnaniarti,

We’re pleased to inform you that your manuscript has been judged scientifically suitable for publication and will be formally accepted for publication once it meets all outstanding technical requirements.

Kind regards,

Isaac Akintoyese Oyekola

Academic Editor

PLOS ONE

Additional Editor Comments (optional):

Reviewers' comments:

Reviewer's Responses to Questions

**Comments to the Author**

1. If the authors have adequately addressed your comments raised in a previous round of review and you feel that this manuscript is now acceptable for publication, you may indicate that here to bypass the “Comments to the Author” section, enter your conflict of interest statement in the “Confidential to Editor” section, and submit your "Accept" recommendation.

Reviewer #3: All comments have been addressed

2. Is the manuscript technically sound, and do the data support the conclusions?

Reviewer #3: Yes

3. Has the statistical analysis been performed appropriately and rigorously? 

Reviewer #3: Yes

4. Have the authors made all data underlying the findings in their manuscript fully available?

Reviewer #3: Yes

5. Is the manuscript presented in an intelligible fashion and written in standard English?

Reviewer #3: Yes

6. Review Comments to the Author

Reviewer #3: table 2 needs clarification. it is still unclear. It is good authors try to justify the result. better author rethink to present the table 2

7. PLOS authors have the option to publish the peer review history of their article (what does this mean? ). If published, this will include your full peer review and any attached files.

**Do you want your identity to be public for this peer review?** For information about this choice, including consent withdrawal, please see our Privacy Policy .

Reviewer #3: **Yes: ** Siti Isfandari

---

## [Editor Report · Acceptance letter]

PONE-D-24-21104R4

PLOS ONE

Dear Dr. Misnaniarti,

I'm pleased to inform you that your manuscript has been deemed suitable for publication in PLOS ONE. Congratulations! Your manuscript is now being handed over to our production team.

Kind regards,

on behalf of

Dr. Isaac Akintoyese Oyekola

Academic Editor

PLOS ONE